# Foodborne acute gastroenteritis outbreak associated with a rare norovirus recombinant GII.10[P16] genotype, Brazil, 2023

**Fernanda Marcicano Burlandy**[1,+]**, Fábio Correia Malta**[1,2]**, Mateus de Souza Mello**[1]**, Alexandre Madi Fialho**[1]**, Gabriel Assad Baduy**[1]**, Ronise Valéria Guarnier**[3]**, Kely Cristiane Souto Moreira**[3]**, Tulio Machado Fumian**[1]

[1]Fundação Oswaldo Cruz-Fiocruz, Instituto Oswaldo Cruz, Laboratório de Virologia Comparada e Ambiental, Rio de Janeiro, RJ, Brasil
[2]Universidade Federal do Rio de Janeiro, Faculdade de Medicina, Hospital Universitário Clementino Fraga Filho, Departamento de Doenças Infecciosas e Parasitárias, Rio de Janeiro, RJ, Brasil
[3]Secretaria de Saúde do Estado do Espírito Santo-SESA, Vigilância Epidemiológica de Doenças de Transmissão Hídrica e Alimentar, Vitória, ES, Brasil

**BACKGROUND** Norovirus is a major cause of acute gastroenteritis (AGE) outbreaks worldwide. On 1 August 2023, the health surveillance agency of Espírito Santo received a notification of a set cases of AGE from patients who attended an event organised by the Municipal Health Department. A local catering company provided lunch on 30 July (30 lunch boxes). The menu provided included macaroni, rice, *tropeiro* beans and cooked potatoes. The ingredients used in the preparation of *tropeiro* beans were beans, banana, cabbage, sausage and cassava flour. Diarrhoea, nausea, vomiting, abdominal pain, headache and weakness were the main reported symptoms peaking between 30 and 31 July.

**OBJECTIVES** In the present study, we aimed to establish the agent responsible for a gastroenteritis outbreak during a lunch event in Espírito Santo State, Brazil.

**METHODS** Stool samples (n = 5) of AGE patients were analysed by real-time quantitative polymerase chain reaction (RT-qPCR) to detect rotavirus A (RVA) and norovirus GI/GII. For norovirus molecular characterisation an open reading frame (ORF)1-2 junction region was used.

**FINDINGS** All samples tested positive for norovirus GII, showing high viral loads. Rotavirus and norovirus GI were not detected in any of the samples. Norovirus sequencing identified a rare recombinant genotype GII.10[P16] as the cause of the outbreak. Norovirus sequences from specimens from five individuals shared 100% of nucleotide (nt) identity and had the highest nt similarity with a South Africa GII.10 strain detected in 2020.

**MAIN CONCLUSION** This is the first report of a rare GII.10[P16] recombinant norovirus strain in Brazil.

Key words: acute diarrhoeal disease - outbreak - norovirus - food handler - Espírito Santo - Brazil

Noroviruses are non-enveloped, single-stranded RNA viruses that are recognised as a leading cause of acute gastroenteritis (AGE). Noroviruses belong to the genus *Norovirus* within the family *Caliciviridae*. Norovirus classification is based on the amino acid similarity of the major capsid protein (VP1), encoded by open reading frame 2 (ORF2). Currently, noroviruses are classified into 10 genogroups (GI-GX) and over 48 genotypes, with the following distribution: nine genotypes in GI, 26 in GII, three in GIII, two each in GIV and GV, two in GVI, and single one each in GVII, GVIII, GIX, and GX.[1,2] Among these, only GI, GII, GIV, GVIII, and GIX are known to infect humans. Additionally, ORF1 encodes the viral non-structural proteins, including the polymerase, which is further categorised into over 60 polymerase types (P-types) based on nucleotide diversity of the RdRP gene (NS7). Norovirus classification has been updated to incorporate dual typing, which includes both the capsid protein genotype (VP1) and the polymerase type (P-type).[1,2]

Noroviruses are highly contagious, and transmission usually occurs through the faecal-oral route or through contaminated food and surfaces. The most common symptoms in infected symptomatic individuals are vomiting, diarrhoea and abdominal pain, beginning between 24 and 48 h after exposure. Although the disease is gen-

Financial support: CNPq (402243/2023-4), FAPERJ [Processo SEI 260003/000530/2023 - Ref. Proc. No. 200.171/2023 - Jovem Cientista do Nosso Estado (TMF); Processo SEI 260003/005914/2024 - Ref. Proc. No. 210.384/2024 (FMB)], PAEF-3/IOC/FIOCRUZ and CNPq PROEP/IOC 441653/2024-3 from the Oswaldo Cruz Institute, Fiocruz. Further support was provided by CGLab, Brazilian Ministry of Health.
TMF is a Productivity Fellowship Researcher from the CNPq.
+ Corresponding author: fburlandy@ioc.fiocruz.br / feburlandy@gmail.com
https://orcid.org/0000-0001-9933-7161

erally self-limiting, symptoms can persist for days, and with the virus's high infectivity can easily lead to widespread illness and outbreaks.[3]

While several genogroups can cause human disease, including GI, GII, GIV, GVIII, and GIX, the majority of norovirus AGE cases are attributed to GI and GII genotypes.[1,2] Among these genotypes, the GII.4 viruses are more often spread via person-to-person contact and is identified as the predominant cause of norovirus infections worldwide.[1,2,4] In a study conducted in the United States between 2009 and 2013, non-GII.4 viruses, such as GI.3, GI.6, GI.7, GII.3, GII.6, and GII.12, were identified associated with AGE foodborne outbreaks.[5] However, other rarer genotypes have been implicated in AGE outbreaks in several countries, such as an outbreak occurred in Wisconsin, United States, associated with a rare GIV genotype affecting 53 individuals, including food handlers, and was linked to individually prepared fruit salad.[6] More recently, the rare GI.5[P4] virus was identified as the cause of a large foodborne AGE outbreak that affected 163 patients, including 15 norovirus-confirmed food handlers at a hotel in Murcia, southeast Spain.[7]

In this study, we report an AGE outbreak caused by a rare norovirus GII.10[P16] genotype. On 1 August 2023, the Health Surveillance Service of Espírito Santo State received complaints of AGE illness from patients who had attended a lunch event. Faecal samples from patients were collected by the State Department of Health to conduct an outbreak investigation in order to identify the cause of the outbreak and preventing further cases.

## SUBJECTS AND METHODS

*Ethics statements* - The Laboratory of Comparative and Environmental Virology (LVCA) houses the Regional Rotavirus Reference Laboratory (RRRL) at the Oswaldo Cruz Institute and is part of the national network for AGE surveillance, which is coordinated by the General Coordination of Public Health Laboratories within the Brazilian Ministry of Health (MoH). Samples were manipulated anonymously, and patients' data were maintained securely. This study was approved by the Ethics Committee of the Oswaldo Cruz Foundation (FIOCRUZ), Brazil (Approval number: CAAE 76063123.5.0000.5248) and was conducted according to the guidelines of the Declaration of Helsinki.

*Outbreak description* - According to the Municipal Health Department report, the incident was linked to lunch served on 30 July 2023. At the catering company's request, food samples collected during the investigation were segregated while awaiting instructions from the local health agency. A total of 30 meal boxes prepared by the same catering company were delivered to City Hall employees that day, containing rice, Brazilian-style beans (*feijão tropeiro*), pasta, and fried chicken, while no salads or beverages were served. Food preparation was completed in the morning on the day of the event, with no advance preparation carried out the previous day. The meals were delivered to the consumption site by 12:00. According to interviews with the affected group, 11 attendees and one (1) food handler reported being ill,

and the first case was reported on 30 July 2023. Diarrhoea, nausea, vomiting, abdominal pain, headache, and weakness were the main reported symptoms. The onset of symptoms began on the same day as the consumption and lasted until the next day. The ages of patients ranged between 30 and 50 years old, and no hospitalisations were reported. The Regional Department of Public Health, in cooperation with the Regional Food Safety Authority, performed an investigation to identify the source. During the facility inspection and interviews with catering staff, one food handler reported experiencing diarrhoea, abdominal pain, and vomiting the night before the event, yet still attended work while symptomatic with vomiting.

*Clinical samples* - Stool specimens (n = 5), which were collected from individuals who attended the event and presented AGE symptoms the day after lunch, were forwarded to LVCA for enteric virus investigation, manipulated anonymously, and patients' data were maintained securely.

*Nucleic acid extraction* - Nucleic acid was extracted from 140 µL of clarified stool suspensions (10% *w/v*) using the automated QIAcube platform using the QIAamp Viral RNA Mini kit (both from QIAGEN, Valencia, CA, USA), following the manufacturer's instructions. Nucleic acid was eluted in 60 µL of elution buffer (AVE) and immediately stored at -80ºC until the molecular analysis. RNase/DNase-free water was used as a negative control in each extraction procedure.

*Enteric viruses' detection and quantification* - All samples were tested for rotavirus A (RVA) and norovirus GI and GII by using TaqMan®-based real-time quantitative polymerase chain reaction (RT-qPCR), with primers and probes as previously described by Zeng et al., Hill et al. and Kageyama et al., respectively.[8,9,10] Primers (COG1F/R; COG2F/R) and probes (RING1C and RING2) targeting ORF1-2 overlapping region were used to detect norovirus GI and GII, respectively. For RVA, primers (NSP3F/R) and a probe targeting the conserved NSP3 gene were used.

All qPCR reactions were carried out in an ABI PRISM 7500® Real-Time System v2.0 (Applied Biosystems, Foster City, CA, USA) using 12.5 µL of SuperScript™ III Platinum® One-Step Quantitative RT-PCR kit (Invitrogen, Carlsbad, CA, USA) and 5 µL of extracted nucleic acid for a final volume of 25 µL. The thermal cycling conditions were carried out as follows: reverse transcription step at 50ºC for 60 min, an initial denaturation step at 95ºC for 5 min and 35 cycles of PCR amplification at 95ºC for 15 s and 60ºC for 1 min. For RT-qPCR protocols, positive, negative, and non-template controls (NTC) were included. Samples with a Ct value ≤ 33 and displaying a characteristic sigmoid curve were considered positive. Each run also included a standard curve with serial dilutions ($10^1$-$10^7$) of double-stranded DNA fragments (gBlock® Gene Fragment, Integrated DNA Technologies, Iowa, USA) containing the target region, to ensure the correct interpretation of the results throughout the study. Viral loads were expressed as genome copies per gram (GC/g) of stool.

*Norovirus sequencing and phylogenetic analysis* - Norovirus GII molecular characterisation by conventional RT-PCR was performed using previously published primers (Mon431 and G2SKR) targeting the 3'end of ORF1 and 5'end of ORF2, with an expected PCR amplicon size of approximately 557 base pairs.[11] Sanger sequencing was performed using both forward and reverse primers with the BigDye™ Terminator v. 3.1 Cycle Sequencing Kit (Applied Biosystems), and reactions were run at FIOCRUZ Institutional Sequencing Platform (PDTIS) on an ABI Prism 3730*xl* Genetic Analyzer (Applied Biosystems).

Chromatogram analysis and consensus sequences were obtained using Geneious Prime 2021.1.1 software (Biomatters Ltd., Auckland, New Zealand). Sequences were genotyped using the Norovirus Genotyping Tool (https://www.rivm.nl/mpf/typingtool/norovirus). Phylogenetic trees were constructed using the maximum-likelihood method and the Kimura two-parameter model (2,000 bootstrap replications for branch support) in MEGA X, with norovirus reference sequences obtained from the National Centre for Biotechnology Information (NCBI) database.[12] Norovirus nucleotide sequences generated in this study were submitted to GenBank and assigned the following accession numbers: PV000908-PV000912.

## RESULTS

All five specimens from the AGE outbreak that were received at our laboratory tested positive for norovirus genogroup II. All specimens had a high viral load ranging from $2.7 \times 10^5$ to $2.11 \times 10^8$ GC/g of stool samples (Ct values of 21.3 to 31.1), with a median of $1.25 \times 10^8$ GC/g (median Ct of 21.8), and patients' ages varied from 37-57 years (Table). RVA was not co-detected in any sample. All stool samples were collected between one and three days after the onset of symptoms. Patients recorded four to six diarrhoea episodes per day, lasting between one and three days. Nausea, vomiting, abdominal pain, headache, and weakness were the other main reported symptoms. Bacteriological tests carried out on lunch boxes (rice, *tropeiro* beans and macaroni) by Central Public Health Laboratory of the Espírito Santo State returned negative results (data not shown).

To investigate the norovirus genotype responsible for the AGE outbreak, we sequenced all five samples targeting the ORF1/ORF2 overlapping region. Molecular characterisation of this region (partial RdRp and capsid) identified the GII.10[P16] in all samples and all nucleotide sequences were 100% identical. In the partial capsid region, our GII.10 sequences showed 100% nucleotide identity with a Russian GII.10[P16] strain (OR478070), detected in August 2022. Furthermore, our sequences showed high nucleotide similarities (> 99% of nucleotide identity) with GII.10[P16] sequences from South Africa in 2020 (OP297036 and OP297037), from the United States detected between 2020 and 2022 (OP609644, OP609645, OP686903, and OP691294), from Australia in 2020 (PP733448), and a GII.10 sequences detected in New Zealand and Spain in 2019 (PP754178 and MT501827). Older sequences from Burkina Faso (JX416402), New Zealand (KT151035), Morocco (KY200646) and Bangladesh (OP609646) were less genetically related, including the unique sequence of GII.10 genotype from Brazil detected in 1993 (KX722403) (Figure A-B).

## DISCUSSION

Noroviruses are the leading cause of foodborne illnesses, with an estimated 5.46 million cases annually.[13] The US Centres of Disease Control and Prevention (CDC) defines a confirmed norovirus outbreak as an occurrence of two or more similar illnesses resulting from a common exposure that is either suspected or laboratory-confirmed to be caused by norovirus.[13,14] Contamination of food can occur at any point in the food supply chain, from production, harvesting, to transportation or handling. In developed countries, contaminated produce has been associated with most norovirus outbreaks in Europe, USA and Asia.[15] However, data from developing countries are still scarce.[16,17] In 2016, an AGE outbreak in Minnesota, United States of America (USA), was traced back to frozen raspberries imported from China. This incident marked the first reported outbreak in the United States associated with commercially distributed frozen berries.[18] Similarly, frozen raspberries contaminated with multiple norovirus genotypes were linked to outbreaks in Quebec, Canada.[19] In a previous

TABLE

Clinical and demographic characteristics of patients associated with a norovirus outbreak, Espírito Santo, Brazil

|  | Patient A | Patient B | Patient C | Patient D | Patient E |
|---|---|---|---|---|---|
| Age / Gender | 37/F | 43/M | 37/F | 44/M | 57/M |
| Days of sample collection after symptoms | 3 | 3 | 1 | 3 | 2 |
| No. of vomiting episode per day (No. of days) | 1(2) | 2 (1) | 5 (1) | * | * |
| Diarrhea episodes per day (total days) | 6 (3) | 4 (3) | 4 (1) | 5 (3) | * |
| Fever (≥ 37.5ºC) | No | No | Yes | No | Yes |
| NoV load (genome copies per gram of stool) | 2,70E+05 | 9,33E+07 | 1,50E+08 | 1,72E+08 | 2,11E+08 |

*Data not informed.

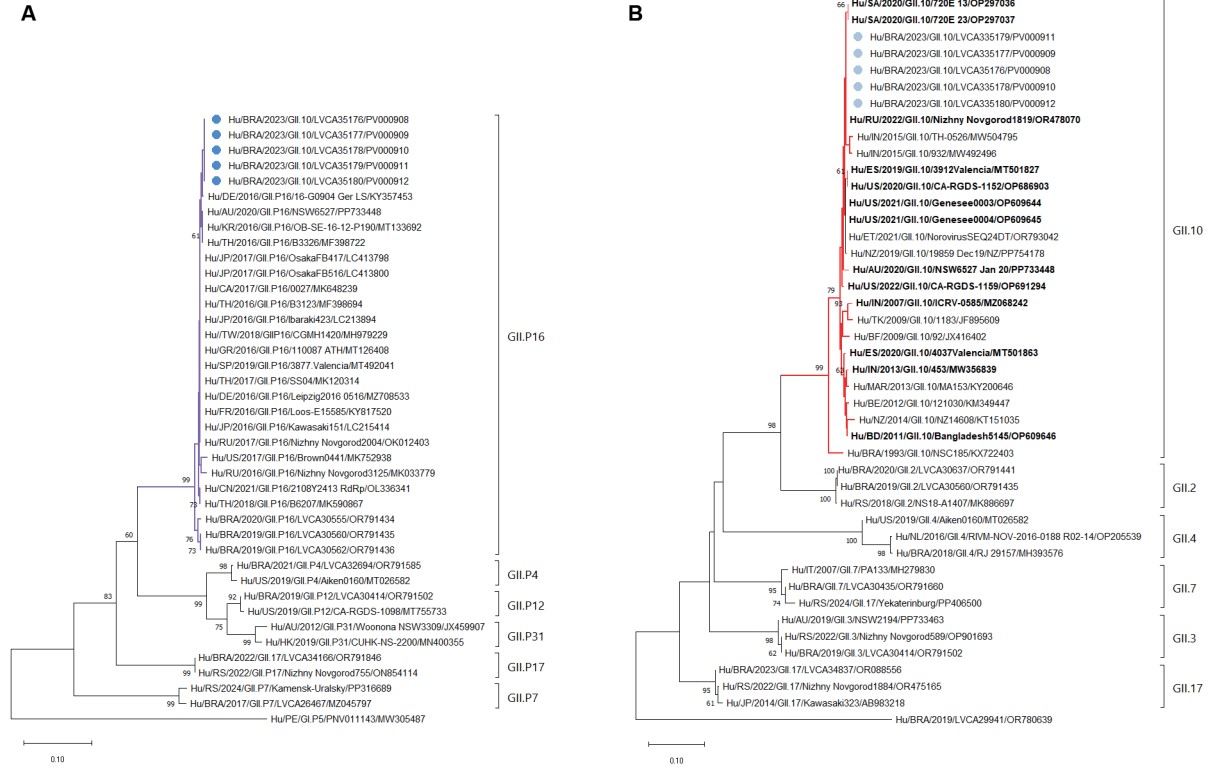

Phylogenetic analysis of norovirus GII strains isolated from Brazilian patients during the outbreak. Phylogenetic tree based on the RdRp nucleotide sequence (A). Phylogenetic tree based on the open reading frame (ORF) 2 nucleotide sequence (B). References strains were downloaded from GenBank and labeled with their respective accession numbers. Sequences obtained from our study (marked with a circle) are shown as per country followed by year of collection, genotype, Laboratory of Comparative and Environmental Virology (LVCA) internal register number and respective accession number (i.e., BRA/2023/GII.10/LVCA35178/PV000908). Neighbor-joining phylogenetic trees were constructed with MEGA-X software and bootstrap tests (2000 replicates), based on the Kimura two-parameter model. Only bootstrap values of > 60 are shown.

study conducted by our group, we identified GII.12 norovirus in stool specimens and ice pop samples during a norovirus outbreak that affected hundreds of people in southern Brazil, 2020.[20]

The epidemiological investigation report revealed that a kitchen assistant (food worker) involved in food preparation, along with her husband (not employed at the facility), had experienced symptoms of vomiting and diarrhoea beginning three days prior to the suspected contamination event. Notably, the food worker had still been symptomatic while working the day before the outbreak but reportedly did not handle the meals in question on the event day, as she was absent. Her symptoms had resolved the day before the outbreak, whereas her husband remained symptomatic on the event day.

Food products may become contaminated with norovirus through exposure to human faeces or vomitus, primarily via infected food handlers who may be either symptomatic or asymptomatic. Epidemiological data indicate that, when contamination sources are identified in foodborne norovirus outbreaks, over 50% of cases originate from infected food service workers.[21,22,23] Recently, Chew et al. reported a foodborne norovirus outbreak linked to six events and a single caterer in Canberra, Australia. In that outbreak, seven attendees tested positive for norovirus, and whilst no food handlers reported illness, one reported their child had recent gastroenteritis.[24]

It is well established that norovirus shedding can persist for days and even weeks.[25] So, in the present study, the source of the contamination was most likely the infected food handler at the catering business. This is of importance for catering businesses that must enforce strict hand hygiene, adhere to proper food handling protocols, and exclusion of symptomatic food handlers to prevent norovirus outbreak.

Regarding worldwide circulation of this recombinant genotype, data from the international norovirus molecular surveillance database (NoroSurv) collected in 16 countries across six continents during 2016-2020 did not detect this recombinant genotype, demonstrating its very limited circulation.[26] In Brazil, the genotype GII.10, combined with P12 (GII.10[P12]), was reported from a single clinical sample collected in 1993.[27] More recently, this recombinant genotype (GII.10[P16]) was detected causing AGE outbreaks in Australia and New Zealand[28] and Spain.[29]

Since 2016, recombinant strains carrying GII.P16, particularly GII.4 and GII.2, have accounted for the majority of norovirus infections worldwide.[30,31,32] Previous studies have suggested that amino acid substitutions in the GII.P16 polymerase may contribute to the emergence of GII.2[P16] recombinants by altering polymerase function.[33,34] Therefore, mutations in the polymerase-encoding region could potentially affect

the enzyme's kinetics and fidelity, indicating that the acquisition of the P16 polymerase might enhance viral fitness for specific genotypes.[33,35]

Our study has some limitations. First, no food items were available for norovirus testing. Second, no stool sample was collected from the symptomatic food handler who reported being ill before the lunch event. Consequently, we were unable to confirm a direct match with the GII.10 found in the five specimens from the patients who got sick. Finally, we did not assess secondary attack rates, as no follow-up interviews or epidemiological investigations were conducted.

This case report describes an AGE outbreak caused by a rare recombinant strain of norovirus GII.10[P16], which has not been reported previously in Brazil. Epidemiologically, food consumed by the lunch participants was likely contaminated by a food worker who had reported diarrhoea and vomiting two days before the event. The features of this outbreak highlight the risk of contaminating food during preparation or handling of food. These findings underscore the critical importance of excluding symptomatic food handlers from food preparation and implementing appropriate disinfection practices including hand washing and surface disinfection.

## ACKNOWLEDGEMENTS

To Rosane Maria Santos de Assis and Sergio Mouta for their technical assistance as well as the staff at the Central Public Health Laboratory, Vitória, Espírito Santo, Brazil, for their technical support.

## AUTHORS' CONTRIBUTION

Conceptualisation, writing-original draft and funding acquisition - FMB and TMF; methodology - FCM, MSM, AMF, GAB and FMB; writing-review & editing - FCM, MSM, RVG, KCSM and TMF; supervision - TMF. All authors have read and agreed to the published version of the manuscript. The authors declare no conflict of interest. Data will be made available on request.

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

# OPEN PEER REVIEW

Memórias do IOC thanks the anonymous reviewers for their contribution to the peer review of this work.

## FIRST REVIEW ROUND

### REVIEWERS COMMENTS

### REVIEWER #1

This manuscript needs significant English editing which I did and I made a Word document with my revisions in track changes mode and I made the document anonymous. Please let me know to whom to email my revision file as this webportal doesn't allow me to upload the Word file.

### REVIEWER #2

a) The abstract is suitable for the manuscript
b) The novelty of the work is the first detection of norovirus GII.10[P16] in Brazil.
c) The methodology applied is appropriate for the detection and description of noroviruses. The results and discussion are clear and concise.
Additional comments:
-Summary: The event dates do not match the description in the Materials and Methods section.
-Introduction: Add more information on norovirus nomenclature, genogroups (G), genotypes and types
-Line 57: Remove the )

### AUTHORS' RESPONSE TO THE REVIEWERS

Dr. Mariza Morgado
Handling Editor
Memórias do Instituto Oswaldo Cruz

May 28th, 2025
Dear Mariza Morgado,
We are forwarding the revised manuscript titled "Foodborne acute gastroenteritis outbreak associated with a rare norovirus recombinant GII.10[P16] genotype, Brazil, 2023.", including the considerations proposed by the reviewers.
All comments/questions and suggestions were judiciously made, as follows below, through which we hope to have fully answered all questions raised so that we might submit the publication to this respectable scientific journal. We would like to thank the reviewers for the valuable suggestions, which considerably enriched this manuscript.
We also take this opportunity to congratulate this journal and editorial team and place ourselves fully at our disposal for any further clarifications.
Sincerely,
Burlandy and coauthors

### RESPONSES TO THE COMMENTS FROM REVIEWERS

Dear Reviewers,
We appreciate all valuable comments, questions, suggestions, and corrections sent. The authors accepted all the suggestions, and we are at your disposal to clarify any doubts that may remain.

Reviewer #1
This manuscript needs significant English editing which I did and I made a Word document with my revisions in track changes mode and I made the document anonymous. Please let me know to whom to email my revision file as this web portal doesn't allow me to upload the Word file.
Comments:
We really appreciate all valuable comments and English editing. We accept all of them and below we added answers about all the appointments.
Question 1. Line 40 - How many?

Answer: We received and processed five stool samples. Number five was included in the text (line 40).

Question 2. Line 142: Ring 1C was not reported by Kageyama et al., 2003 - Only ring 1a and ring 1b.

Answer: We included the correct reference (Hill et al., 2010).

Question 3. Lines 150-152: PCR reaction conditions are missing here.

Answer: The appropriate information was included ("The thermal cycling conditions were carried out as follows: RT step at 50ºC for 60 min, an initial denaturation step at 95◦C for 5 min and 35 cycles of PCR amplification at 95ºC for 15 s and 60ºC for 1 min.") in lines 146-148.

Question 4. Line 164: Please use appropriate reference.

Answer: The correct reference Chhabra et al., 2021 (11) was included.

Question 5. Line 179

Answer: We agree with using specimens than samples. The correction was made (lines 179-182).

Question 6. Line 202-211

Answer: We did adjustments in results text as pointed.

Question 6. Line 231-236

Answer: We did adjustments in discussion text as pointed.

Reviewer #2

a) The abstract is suitable for the manuscript

b) The novelty of the work is the first detection of norovirus GII.10[P16] in Brazil.

c) The methodology applied is appropriate for the detection and description of noroviruses. The results and discussion are clear and concise.

Additional comments:

-Summary: The event dates do not match the description in the Materials and Methods section.

-Introduction: Add more information on norovirus nomenclature, genogroups (G), genotypes and types.

-Line 57: Remove the )

Comments:

Question 1. Summary: The event dates do not match the description in the Materials and Methods section.

Answer: The correct date, July 30 (line 40), was written.

Question 2. Introduction: Add more information on norovirus nomenclature, genogroups (G), genotypes and types.

Answer: More information was added (lines 59-68).

Question 3. Line 57: Remove the )

Answer: The appropriate correction text was made in line 67.

## SECOND REVIEW ROUND

### REVIEWERS COMMENTS

### REVIEWER #1

Reviewer comments: The authors conducted a detailed review of the manuscript, taking into account the comments. I think the work has improved significantly.

### REVIEWER #2

Reviewer comments: no comments.

