## [Reviewer Report · FIRST REVIEW ROUND - REVIEWERS' COMMENTS]

## Reviewer #1

This manuscript needs significant English editing which I did and I made a Word document with my revisions in track changes mode and I made the document anonymous. Please let me know to whom to email my revision file as this web portal doesn't allow me to upload the Word file.

## Reviewer #2

a) The abstract is suitable for the manuscript

b) The novelty of the work is the first detection of norovirus GII.10[P16] in Brazil.

c) The methodology applied is appropriate for the detection and description of noroviruses. The results and discussion are clear and concise.

## Additional comments

- Summary: The event dates do not match the description in the Materials and Methods section.

- Introduction: Add more information on norovirus nomenclature, genogroups (G), genotypes and types.

- Line 57: Remove the )

## AUTHORS' RESPONSE TO THE REVIEWERS

Dr. Mariza Morgado

Handling Editor

Memórias do Instituto Oswaldo Cruz

May 28th, 2025

Dear Mariza Morgado,

We are forwarding the revised manuscript titled “Foodborne acute gastroenteritis outbreak associated with a rare norovirus recombinant GII.10[P16] genotype, Brazil, 2023.”, including the considerations proposed by the reviewers.

All comments/questions and suggestions were judiciously made, as follows below, through which we hope to have fully answered all questions raised so that we might submit the publication to this respectable scientific journal. We would like to thank the reviewers for the valuable suggestions, which considerably enriched this manuscript.

We also take this opportunity to congratulate this journal and editorial team and place ourselves fully at our disposal for any further clarifications.

Sincerely,

Burlandy and coauthors

## RESPONSES TO THE COMMENTS FROM REVIEWERS

Dear Reviewers,

We appreciate all valuable comments, questions, suggestions, and corrections sent. The authors accepted all the suggestions, and we are at your disposal to clarify any doubts that may remain.

## Reviewer #1 - Authors' responses

We really appreciate all valuable comments and English editing. We accept all of them and below we added answers about all the appointments.

Question 1. Line 40 - How many?

Answer: We received and processed five stool samples. Number five was included in the text (line 40).

Question 2. Line 142: Ring 1C was not reported by Kageyama et al., 2003 - Only ring 1a and ring 1b.

Answer: We included the correct reference (Hill et al., 2010).

Question 3. Lines 150-152: PCR reaction conditions are missing here.

Answer: The appropriate information was included (“The thermal cycling conditions were carried out as follows: RT step at 50°C for 60 min, an initial denaturation step at 95°C for 5 min and 35 cycles of PCR amplification at 95°C for 15 s and 60°C for 1 min.”) in lines 146-148.

Question 4. Line 164: Please use appropriate reference.

Answer: The correct reference Chhabra et al., 2021 (11) was included.

Question 5. Line 179

Answer: We agree with using “specimens” rather than “samples.” The correction was made (lines 179-182).

Question 6. Lines 202-211

Answer: We did adjustments in results text as pointed.

Question 7. Lines 231-236

Answer: We did adjustments in discussion text as pointed.

## Reviewer #2 - Authors' responses

Question 1. Summary: The event dates do not match the description in the Materials and Methods section.

Answer: The correct date, July 30 (line 40), was written.

Question 2. Introduction: Add more information on norovirus nomenclature, genogroups (G), genotypes and types.

Answer: More information was added (lines 59-68).

Question 3. Line 57: Remove the )

Answer: The appropriate correction text was made in line 67.

---

## [Reviewer Report · REVIEWERS' COMMENTS]

## Reviewer #1

Reviewer comments: The authors conducted a detailed review of the manuscript, taking into account the comments. I think the work has improved significantly.

## Reviewer #2